# Activation of Endothelial Large Conductance Potassium Channels Protects against TNF-α-Induced Inflammation

**DOI:** 10.3390/ijms24044087

**Published:** 2023-02-17

**Authors:** Tatiana Zyrianova, Kathlyn Zou, Benjamin Lopez, Andy Liao, Charles Gu, Riccardo Olcese, Andreas Schwingshackl

**Affiliations:** 1Departments of Pediatrics, University of California Los Angeles, Los Angeles, CA 90095, USA; 2Departments of Anesthesiology and Perioperative Medicine, University of California Los Angeles, Los Angeles, CA 90095, USA; 3Departments of Physiology, University of California Los Angeles, Los Angeles, CA 90095, USA

**Keywords:** lung inflammation, TNF-α, large conductance K^+^ channels, L-type voltage-gated Ca^2+^ channels, Nifedipine, NS1619, endothelial cells, CCL-2, IL-6, pathway analysis

## Abstract

Elevated TNF-α levels in serum and broncho-alveolar lavage fluid of acute lung injury patients correlate with mortality rates. We hypothesized that pharmacological plasma membrane potential (Em) hyperpolarization protects against TNF-α-induced CCL-2 and IL-6 secretion from human pulmonary endothelial cells through inhibition of inflammatory Ca^2+^-dependent MAPK pathways. Since the role of Ca^2+^ influx in TNF-α-mediated inflammation remains poorly understood, we explored the role of L-type voltage-gated Ca^2+^ (Ca_V_) channels in TNF-α-induced CCL-2 and IL-6 secretion from human pulmonary endothelial cells. The Ca_V_ channel blocker, Nifedipine, decreased both CCL-2 and IL-6 secretion, suggesting that a fraction of Ca_V_ channels is open at the significantly depolarized resting Em of human microvascular pulmonary endothelial cells (−6 ± 1.9 mV), as shown by whole-cell patch-clamp measurements. To further explore the role of Ca_V_ channels in cytokine secretion, we demonstrated that the beneficial effects of Nifedipine could also be achieved by Em hyperpolarization via the pharmacological activation of large conductance K^+^ (BK) channels with NS1619, which elicited a similar decrease in CCL-2 but not IL-6 secretion. Using functional gene enrichment analysis tools, we predicted and validated that known Ca^2+^-dependent kinases, JNK-1/2 and p38, are the most likely pathways to mediate the decrease in CCL-2 secretion.

## 1. Introduction

TNF-α is a key regulatory molecule in the development of both direct (e.g., pneumonia) and indirect (e.g., non-pulmonary sepsis) phenotypes of acute lung injury (ALI) [1,2]. In inflammatory and immune cells, such as macrophages and monocytes, the signaling cascades activated by TNF-α are fairly well-established [3]. Much less is known about the mechanisms underlying TNF-α signaling in lung resident cells, such as vascular endothelial and alveolar epithelial cells, which are increasingly being recognized as important contributors to inflammatory mediator secretion and alveolar barrier dysfunction during ALI [4].

The binding of TNF-α to its main receptor, TNF-α receptor 1 (TNFR1), activates multiple inflammatory signaling cascades simultaneously, which ultimately promote inflammatory mediator secretion, barrier dysfunction, and cell apoptosis [1]. In macrophages and airway smooth muscle cells, TNF-α-induced cell activation is associated with an increase in intracellular Ca^2+^ (iCa^2+^) levels [5,6], but during ALI, the importance and downstream consequences of Ca^2+^ flux across the alveolar–capillary interface remains poorly understood. To complicate matters, in alveolar endothelial and epithelial cells, both Ca^2+^-dependent and -independent secretory mechanisms have been observed. Specifically, in vascular endothelial cells, some investigators found an increase in iCa^2+^ levels following TNF-α treatment [7,8], while others observed no effect of TNF-α on endothelial iCa^2+^ concentrations [9]. Recent data obtained in human umbilical vein endothelial cells suggest that TNF-α-mediated stimulus-secretion coupling, including NF-kB-1 activation and subsequent proinflammatory IL-1β secretion, is a Ca^2+^ influx-dependent process [10], but the relevance of these findings to the development of ALI remains unknown. In alveolar epithelial cells, we previously showed that the secretion of the inflammatory cytokine IL-6 occurs independently of intra- and extracellular Ca^2+^ concentrations, whereas the secretion of CCL-2 (MCP-1) relies on the mobilization of iCa^2+^ stores but not extracellular Ca^2+^ levels [11] in these cells. Despite these differences in secretory mechanisms, both IL-6 and CCL-2 are consistently elevated in the serum and broncho-alveolar lavage (BAL) fluid of patients with ALI/ARDS and correlate with patient mortality rates [12,13].

These knowledge gaps highlight that a more detailed understanding of the electro-chemical signaling mechanisms occurring at the alveolar–capillary interface is crucial for the design of new therapeutic strategies against ALI. Specifically, we face an urgent need to clarify (i) the role of L-type voltage-gated Ca^2+^ (Ca_V_) channels in regulating Ca^2+^ influx and inflammatory cytokine secretion during ALI, (ii) the importance of resting plasma membrane potential (Em) alterations in regulating alveolar iCa^2+^ homeostasis, and (iii) the protein–protein interaction (PPI) networks that translate Em and iCa^2+^ alterations into inflammatory cytokine secretion. Therefore, this study was designed to (a) determine the role of Ca_V_ channels in TNF-α-induced CCL-2 and IL-6 secretion from human pulmonary endothelial cells, two key inflammatory cytokines involved in the development of ALI [12,13], (b) explore the potential of counteracting Ca^2+^ influx-dependent inflammatory cytokine secretion via pharmacological Em hyperpolarization, and (c) establish the intracellular signaling mechanisms and PPI networks that confer such protection. We hypothesized that pharmacological Em hyperpolarization protects against TNF-α-induced CCL-2 and IL-6 secretion from human pulmonary endothelial cells through the inhibition of inflammatory Ca^2+^-dependent MAPK pathways.

Our results in cultured and primary human pulmonary endothelial cells show that TNF-α stimulation increases inflammatory CCL-2 and IL-6 secretion from these cells, which can be inhibited by blocking L-type voltage-gated Ca^2+^ (Ca_V_) channels. Alternatively, a similar protective effect on CCL-2 secretion can be achieved by inhibiting Ca_V_ activity via large conductance K^+^ (BK) channel-mediated Em hyperpolarization in human pulmonary endothelial cells, which results in the downstream disruption of known Ca^2+^-dependent JNK-1/2 and p38 MAPK signaling pathways.

## 2. Results

### 2.1. L-Type Ca_V_ Channels Regulate TNF-α-Induced Inflammatory Cytokine Secretion from Human Pulmonary Endothelial Cells

TNF-α-induced secretory processes are Ca^2+^-dependent in many cell types, but little is known about the role of endothelial Ca^2+^ influx during TNF-α-induced inflammation in the lung. Initial dose–response experiments revealed that 5 ng/mL and 50 ng/mL TNF-α evoked a similar increase in CCL-2 and IL-6 secretion from cultured human pulmonary microvascular endothelial cells (Figure 1A,B and Appendix A) and, therefore, we used the lower 5 ng/mL dose for this study to avoid any potential cell toxicity caused by higher TNF-α doses. In fact, cell viability remained >85% under all reported treatment conditions. Blocking L-type Ca_V_ channels with Nifedipine in these cells inhibited CCL-2 and IL-6 secretion by 76% and 42%, respectively, demonstrating the role of L-type Ca_V_ channels in these TNF-α-induced secretory processes in cultured human pulmonary endothelial cells (HULEC5a). To a much lesser degree, Nifedipine also decreased baseline CCL-2 and IL-6 secretion from unstimulated endothelial cells (by 25% and 38%, respectively).

Since studies on rat alveolar epithelial and calf pulmonary artery endothelial cells reported a significant depolarized Em [14,15], we measured the baseline Em in cultured human pulmonary endothelial cells and found a baseline Em value of −6 ± 1.9 mV.

Based on our findings with Nifedipine, the involvement of Nifedipine-sensitive [16,17] Ca_V_ channels in the regulation of CCL-2 and IL-6 secretion from TNF-α-stimulated cultured endothelial HULEC5a is suggested. We next investigated whether Em hyperpolarization via large conductance K^+^ (BK) channel activation could be used as an alternative approach to counteract Ca_V_-dependent cytokine secretion. First, we used high-throughput, fluorescence-based FLIPR assays to demonstrate that the BK channel opener, NS1619 (30 µM), potently hyperpolarized the endothelial Em in both untreated and TNF-α-stimulated cultured human pulmonary microvascular endothelial cells (Figure 1C,D). Of note, in these assays, Em hyperpolarization is reflected by a decrease in relative fluorescence, whereas an increase in relative fluorescence is indicative of Em depolarization. As an internal positive control for the assay, we artificially raised the extracellular K^+^ concentration with K_2_SO_4_ to 45 mM, which resulted in the expected Em depolarization.

Second, we quantified the baseline and TNF-α-induced CCL-2 and IL-6 secretion from pulmonary endothelial cells after BK channel activation-mediated Em hyperpolarization with NS1619 (Figure 1E,F). Similar to the effects observed with Nifedipine, NS1619 decreased TNF-α-induced CCL-2 secretion from cultured human pulmonary microvascular endothelial (HULEC5a) cells from 4386 ± 128 pg/mL to 2163 ± 434 pg/mL (mean ± SEM; *p* ≤ 0.001; representing a 49% decrease; Figure 1E), and in primary human pulmonary artery endothelial cells (HPAEC) from 133,377 ± 1920 to 86,370 ± 5594 pg/mL (mean ± SEM; *p* ≤ 0.0001; representing a 36% decrease; Figure 1F). In contrast to CCL-2, TNF-α-induced IL-6 release was not affected by BK channel activation (Figure 1G). Furthermore, the physiologic baseline CCL-2 and IL-6 release from unstimulated endothelial cells remained unchanged after BK channel activation. In contrast to pulmonary endothelial cells, in primary human alveolar epithelial cells, another important cell type that is injured during ALI, BK channel activation-mediated Em hyperpolarization with NS1619 had no effect on TNF-α-induced CCL-2 or IL-6 secretion (Appendix A).

Taken together, these results provide supportive evidence for the role of L-type Ca_V_ channels in inflammatory cytokine secretion from human pulmonary endothelial cells. Furthermore, our findings in cultured and primary human pulmonary endothelial cells demonstrate that endothelial Em hyperpolarization via BK channel activation may constitute a novel approach to counteract TNF-α-induced CCL-2 secretion.

### 2.2. BK Channel Activation Decreases Inflammatory CCL-2 Secretion via Inhibition of Ca^2+^-Dependent JNK-1/2 and p38 Signaling Pathways

We used the STRING database to predict the protein–protein interaction (PPI) networks involved in the observed decrease in TNF-α-induced CCL-2 secretion following BK channel activation (NS619). This powerful, free-of-charge platform is one of the most popular approaches for analyzing the associations of proteins and genes obtained from proteomics and other high-throughput technologies, and it contains information from more than 14,000 organisms.

In our prediction model, we included our four experimentally validated (source) proteins from Figure 1 as input variables: TNF-α, L-type Ca_V_ channels, CCL-2, and BK channels. The first step, after setting the confidence of the interaction level to “high” (0.8; scale 0–1), was for the STRING model to create a predictive PPI network containing the following top nine interaction partners: CCR-2, TNFRSF-1B, BIRC-2, TRAF-2, TNFRSF-1A, RIPK-1, TRADD, FADD, IKBKG (Figure 2A). The second step was to use STRING cluster analysis to identify three specific sub-clusters within the initial PPI network (Figure 2B): (i) ion channels (including BK and L-type Ca_V_ channels), (ii) cytokines (including CCL-2 and its receptor CCR-2), and (iii) members of the TNF-α pathway (including TNF-α, TNFRSF-1A, TNFRSF-1B, BIRC-2, TRAF-2, RIPK-1, TRADD, FADD, IKBKG). In this model, the cytokine and TNF-α pathway sub-clusters interacted with each other via the proinflammatory CCL-2/TNF-α axis and their respective receptors, CCR-2 and TNFRSF-1B (dashed lines in Figure 2B), which shared three out of four edges, suggested a high likelihood of interaction. Based on the currently available information, with the same input variables, the ion channel sub-cluster remained isolated from the rest of the PPI network, except for one previous report that linked the BK channels to the cytokine CCL-2 [18], highlighting the substantial lack of knowledge in this area.

After establishing the sub-clusters involved in our prediction model (Figure 2B), in a third step, we used the STRING-KEGG (Kyoto encyclopedia of genes and genomes) pathway analysis to determine the specific signaling pathways connecting the predicted sub-clusters to each other (Figure 2C,D). This approach is based on the prediction score (reflective of strength), false discovery rate (a statistical approach to correct for random events that falsely appear significant), and our current biological knowledge of TNF-α-induced inflammation in the lung. STRING-KEGG analysis revealed the following three pathways as the most likely involved in the BK channel-mediated inhibition of TNF-α-induced CCL-2 secretion: (i) RIG-1, (ii) NF-kB-1, and (iii) MAPK. Interestingly, these pathways overlap across the ion channel, cytokine, and TNF-α sub-clusters identified in Figure 2B. Although simply ranked by strength and a false discovery rate, MAPK pathways are listed lower than several other candidates (Figure 2C). Nevertheless, we included the MAPK group in our validation experiments since MAPK are part of well-established Ca^2+^-dependent inflammatory pathways in numerous published ALI models [18,19]. Based on the STRING-KEGG analysis, we predicted the three most likely targets of BK and Ca_V_ channels which are at the intersection of all three pathways (RIG-1, NF-kB-1, and MAPK, as indicated by three different colors), and are primarily connected with the source protein TNF-α: IKBKG, TRADD, and TRAF2 (Figure 2D).

Based on the information obtained from the PPI prediction models, we experimentally validated the effects of the BK channel opener NS1619 on the activation of RIG-1, NF-kB-1/p65, and MAPK (p38, ERK-1/2, JNK-1/2) pathways in untreated and TNF-α-stimulated cultured human pulmonary microvascular endothelial cells (Figure 3). We found that TNF-α treatment caused phosphorylation and the activation of three key members of the MAPK pathway, namely JNK-1/2, p38, and ERK-1/2, as well as NF-kB-1/p65 phosphorylation (Figure 3A–C,E). The treatment of TNF-α-stimulated endothelial cells with the BK channel opener NS1619 counteracted the phosphorylation and thus the activation of JNK-1/2 and p38 (Figure 3A,B), but not ERK-1/2 or NF-kB-1/p65 (Figure 3C,F). Of note, in unstimulated endothelial cells, NS1619 decreased baseline phosphorylation levels of p38 and ERK-1/2 (Figure 3A,B). In contrast to JNK1/2 and p38 activation, RIG-1 expression levels were not affected by TNF-α (Figure 3D). Additionally, NS1619 had no effect on RIG-1 expression in unstimulated and TNF-α-stimulated cultured endothelial cells (Figure 3D). These findings suggest that JNK-1/2 and p38 pathways are likely responsible for the BK channel-mediated protective effects seen in cultured human pulmonary endothelial cells.

To experimentally validate that BK channel activation reduces CCL-2 secretion by inhibiting the known Ca^2+^-dependent MAPK pathways JNK-1/2- and p38, we blocked these pathways with pharmacological inhibitors and measured CCL-2 secretion. We found that the inhibition of JNK-1/2 (JNK inhibitor VIII) and p38 (SB203580) decreased TNF-α-induced CCL-2 release by 73% (from 4386 ± 128 to 1189 ± 344 pg/mL) and 69% (from 4386 ± 128 to 1381± 280 pg/mL), respectively. The combination of NS1619 with the JNK inhibitor VIII or SB203580 decreased TNF-α-stimulated CCL-2 secretion by 96 % (from 4386 ± 128 to 140 ± 15 pg/mL) and by 97% (4386 ± 128 to 129 ± 20 pg/mL; means ± SEM, *p* ≤ 0.0001; Figure 3G), respectively.

After creating prediction models using the STRING database (Figure 2) and the experimental validation of NS1619-mediated effects on intracellular PPI networks (Figure 3), in the next analysis step, we utilized STITCH software to explore protein–chemical interactions (Figure 4). The STITCH search tool allows specific interactions to be investigated between chemical NS1619 and the intracellular Ca^2+^-dependent MAPK proteins (JNK-1/2 and p38) affected by NS1619 treatment. This database contains 68,000 different chemicals, including 2200 drugs, and connects them to 1.5 million genes across 373 genomes using two main parameters; the “Count in gene set” is a parameter used in functional gene enrichment analysis and is a powerful analytical method for interpreting gene expression data. It ranks all genes in a data set, then calculates an enrichment score for each gene set (the higher the “count”, the more often members of that gene set occur at the top of the ranked data set). The second parameter, “False discovery rate (FDR)”, is based on a statistical approach that is used in high-throughput experiments to correct for random events that falsely appear significant (low FDR = high probability of involvement of a given pathway in the network). As input variables (source proteins) for this model, we entered NS1619 as a chemical, and BK channels (KCNMA1), Ca_V_ channels (CACNA1), TNF-α, CCL-2, JNK-1/2, and p38 as experimentally validated source proteins (Figure 1 and Figure 3). By setting the confidence level of interaction to “high” (0.8, scale 0–1), we identified five additional signaling molecules that interacted with our input variables: JNK-interacting protein-1 (JIP-1), two subunits of the Activator Protein-1 (JUND, JUN), tumor protein 53 (TP53), and the iron-regulated transcriptional activator AFT-2. Furthermore, STITCH-KEGG pathway analysis also revealed that nine out of 11 networks (“node”) proteins were part of the MAPK signaling cascade (green nodes in Figure 4). Although all MAP kinases and transcription factors JUN and AFT-2 are connected by a high number of edges [3,4] within the network, suggesting a high probability of interactions, only JNK-1/2 and p38 directly interacted with TNF-α and CCL-2, thus suggesting a central role for these two targets in the observed BK channel-mediated effects. Since NS1619 is currently not a U.S. Food and Drug Administration (FDA)-approved agent, in this model it has no connections within the cluster.

Altogether, the combination of PPI network modeling (STRING), protein-chemical interaction analysis (STITCH), and experimental phosphorylation and protein expression studies provides strong evidence that the BK channel activation-mediated decrease in TNF-α-induced CCL-2 secretion is caused by the inhibition of Ca^2+^ -dependent JNK-1/2 and p38 signaling pathways.

## 3. Discussion

Very little is known about the electrochemical signaling mechanisms underlying the inflammatory processes occurring at the alveolar–capillary interface during ALI, which makes it challenging to develop new targeted therapeutic approaches against this devastating disease. In particular, the role of Ca^2+^ influx in inflammatory mediator secretion from lung resident (endothelial, epithelial) and immune cells in this highly complex inflammatory environment is poorly understood. In pulmonary endothelial and epithelial cells, the main structural components of the alveolar–capillary interface [20,21], both Ca^2+^-dependent and -independent secretory mechanisms have been observed. Previous studies from our lab revealed that the TNF-α-induced secretion of the inflammatory cytokine IL-6 from cultured alveolar epithelial cells occurred independently of intra- and extracellular Ca^2+^ concentrations. Inflammatory CCL-2 (MCP-1) secretion, on the other hand, relies on the mobilization of intracellular Ca^2+^ (iCa^2+^) stores but not extracellular Ca^2+^ levels in cultured alveolar epithelial cells [11,22]. In vascular endothelial cells, some investigators found an increase in iCa^2+^ levels following TNF-α treatment [11,12], while others observed no effect of TNF-α on iCa^2+^ concentrations [13]. In immortalized human umbilical vein endothelial cells (HUVEC), TNF-α-mediated stimulus-secretion coupling occurred via Ca^2+^ influx-dependent NF-kB-1 activation and subsequent proinflammatory IL-1β release [14], but the relevance of these findings for the development of ALI remains unknown.

Detailed information about Ca^2+^ influx mechanisms at the alveolar–capillary interface is scarce, and therefore, in this study, we focused on human pulmonary endothelial cells. Previous reports showed the expression of transient receptor potential (TRP) channels in lung endothelial [23] and epithelial cells [24], but our knowledge about voltage-gated Ca_V_ channels in the lungs and their potential contribution to the regulation of iCa^2+^ concentrations is limited. One study, using rat lung slices, found the expression of R-, T-, and N-type Ca_V_ channels in smooth muscle cells and nerve fibers and L-type (Ca_V_ 1.2 and 1.3) channels in lung epithelial but not endothelial cells [25]. Therefore, our Nifedipine studies show for the first time the involvement of L-type Ca_V_ channels in inflammatory CCL-2 and IL-6 secretion (Figure 1A,B). As a potential therapeutic option against ALI/ARDS, Ca_V_ channels are of particular interest since they can be blocked by Nifedipine, an already FDA-approved drug. Additionally, our Nifedipine studies also revealed that L-type Ca_V_ channels regulate physiologic baseline CCL-2 and IL-6 secretion from human endothelial cells, although, biologically this may be of lesser importance than the role of L-type Ca_V_ channels in cytokine secretion during TNF-α-induced inflammation. The inhibitory effects of Nifedipine on cytokine secretion from endothelial cells indicate that a fraction of L-type Ca_V_ channels is open, and mediates the Ca^2+^ influx required for CCL-2 and IL-6 secretion (Figure 5A). In organs other than the lung, similar anti-inflammatory effects of Nifedipine have been reported. In human chondrocytes, Nifedipine inhibits oxidative stress and the secretion of inflammatory mediators by activating the nuclear factor erythroid-2-related factor 2 (Nrf2) pathway [26]. Similar anti-inflammatory effects were observed in atherosclerosis and a cuff-induced vascular inflammation model, where Nifedipine inhibited TNF-α-induced reactive oxygen species (ROS) generation and CCL-2 gene expression, but CCL-2 protein levels were not measured in either of these studies [27,28]. Our results expand our knowledge in this field by showing for the first time the anti-inflammatory effects of Nifedipine in TNF-α-stimulated cultured human pulmonary endothelial cells. However, whether these effects hold true in freshly isolated endothelial cells, needs to be established.

In addition to demonstrating that TNF-α-stimulated CCL-2 and IL-6 secretion from pulmonary endothelial cells is dependent on Ca^2+^ influx via Ca_V_ channels, this study also revealed that TNF-α itself does not alter the endothelial Em (Figure 1C). This is not too surprising since Ca^2+^ currents through Ca_V_ channels are relatively small, and iCa^2+^ can rapidly be shifted into intracellular stores or extruded via Ca^2+^ exchange mechanisms [29]. In addition, the lack of effect of TNF-α on the Em makes it unlikely that in endothelial cells TNF-α directly activates large conductance K^+^ (BK) channels, and we found no changes in BK channel expression in TNF-α-treated endothelial cells (Appendix A). Previous studies report varying effects of TNF-α on K^+^ currents and the Em in different cell types. Similar to our results, in nociceptive rat dorsal root ganglion cells, TNF-α had no effect on the activity of voltage-gated K^+^ channels [30], while in rat myenteric neurons, the rat liver cell line (HTC), and in retinal ganglion cells, TNF-α induced Em hyperpolarization [31,32,33].

Given the relatively depolarized Em values measured in our human pulmonary microvascular endothelial cells (−6 ± 1.9 mV), which are consistent with data in the literature on mammalian endothelial cells [15], we next explored whether the inhibitory effects of Ca_V_ inhibition with Nifedipine on inflammatory cytokine secretion could be replicated by inhibiting Ca_V_ activation via Em hyperpolarization [34]. We found that pharmacological Ca_V_ inhibition with Nifedipine decreased both CCL-2 and IL-6 secretion from cultured TNF-α-treated endothelial cells, while Em hyperpolarization with the BK channel opener NS1619 only inhibited CCL-2 but not IL-6 secretion (Figure 5B,C). These findings suggest that even after BK channel-induced Em hyperpolarization, enough residual Ca_V_ channels remained open to allow for IL-6 secretion to occur. This is an important finding since, historically, IL-6 has been viewed as a pro-inflammatory cytokine that was found to be elevated in patients with ALI/ARDS, but recent reports show beneficial effects of IL-6 in lipopolysaccharide (LPS) and mechanical ventilation-induced ALI models [35]. An earlier study suggested a similar protective effect of IL-6 in a hyperoxia-induced ALI model [36]. Therefore, the inhibition of the neutrophil-chemoattractant CCL-2 with NS1619 without impairing IL-6 secretion could potentially be beneficial in certain types of ALI. It is important to note that while Em hyperpolarization with NS1619 had a profound inhibitory effect on TNF-α-stimulated CCL-2 secretion from human pulmonary endothelial cells, this effect was absent in human alveolar epithelial cells (Figure 1 and Appendix A). These findings are consistent with our own previously published study, where we showed that two structurally different BK channel activators (NS1619 and NS19504) showed an identical degree of Em hyperpolarization and a decrease in CCL-2 secretion from LPS-stimulated cultured human pulmonary endothelial cells [18]. However, in the LPS-induced inflammation model, Ca_V_ inhibition with Nifedipine did not affect the CCL-2 levels in human pulmonary endothelial cells, highlighting important differences in the role of Ca_V_ channels between TNF-α- and LPS-mediated inflammation.

To predict the protein–protein interaction (PPI) networks that integrate TNF-α, BK, Ca_V_ channels, and the cytokine CCL-2 in our inflammation model, we used the STRING database. This powerful and free-of-charge web-based resource contains both gene and protein information from numerous sources, including experimental data, computational prediction methods, and public text collections [37]. This predictive model identified the top nine interaction partners within our network and divided them into three sub-clusters (Figure 2A,B). As expected, TNF-α pathway molecules and cytokine sub-clusters connected with each other through well-known relationships between TNF-α/CCL-2 and TNFRSF1B/CCR-2. TNF-α is a pro-inflammatory cytokine that exerts its action through binding to two cognate receptors, TNFRSF-1A, and TNFRSF-1B, and induces CCL-2 cytokine secretion from lung cells [38]. Based on data from our previous study using an LPS-induced ALI model [18], BK channels are connected to the TNF-α cluster through the cytokine CCL-2. While the STRING database is helpful in defining interacting protein clusters, STRING-KEGG analysis can be used to predict specific signaling pathways employed by the different clusters. Using this method, we identified the three most likely pathways within our network that could explain the BK-mediated inhibitory effects on CCL-2 secretion: RIG-1, NF-kB-1, and MAPK (Figure 2C,D). These signaling cascades are well-aligned with literature studies on TNF-α-induced inflammation in primary human alveolar epithelial and endothelial cells [38,39,40]. Our prediction models formed the basis for our experimental validation studies (Figure 3), which indeed confirmed the Ca^2+^-dependent MAPK, JNK-1/2, and p38 as the most likely signaling pathways responsible for the observed reduction in CCL-2 secretion, since the blocking of these pathways with pharmacological inhibitors reduced endothelial CCL-2 secretion (Figure 3G). RIG-1 and NF-kB-1 pathways, on the other hand, are unlikely major contributors to this process. While MAPK signaling is well-known to be a Ca^2+^-dependent process [41], to the best of our knowledge, currently, no reports suggest Ca^2+^-dependency for the RIG-1 axis. This could explain the lack of NS1619-mediated Em hyperpolarization on RIG-1 activation [42,43]. On the other hand, NF-kB-1 activation is generally viewed as Ca^2+^-dependent and, therefore, the lack of NS1619 effect on NF-kB-1 phosphorylation suggests that its activation depends on Ca^2+^ sources other than Ca^2+^ influx through Ca_V_ channels. Additionally, we predicted the three most likely targets of BK and Ca_V_ channels: IKBKG, TRADD, and TRAF2 (Figure 2D). These target molecules are at the intersection of all three pathways (RIG-1, NF-kB-1, and MAPK), as indicated by the three different colors in each node, and interconnected with a source protein TNF-α. In the future, it would be interesting to see if NS1619 downregulates these proteins in endothelial cells.

While the STRING/KEGG databases are useful tools for predicting protein–protein interaction networks and related signaling pathways, for the last reported set of experiments, we employed the STITCH database (Figure 4) to clarify the relationships between the chemical, NS1619, and the protein clusters identified in Figure 2. STITCH is another powerful, free-of-charge database that predicts chemical/drug-to-protein interactions by integrating information about metabolic pathways, crystal structures, drug binding experiments, and drug–target relationships. STITCH connects proteins from 630 organisms to over 74,000 different chemicals, including 2200 drugs [44]. Using this tool, we identified the transcription factor JUN as a central component of NS1619-mediated effects on TNF-α-induced CCL-2 secretion (Figure 4 and Figure 5) since (i) JUN associates with TNF-α and CCL-2 simultaneously, (ii) in lung resident cells, JUN (but not TP53 or JUND) primarily targets JNK-1 [45], which in our hands is inhibited by NS1619 treatment (Figure 3), and (iii) JUN-interacting protein-1 (JIP-1) is known to be required for JNK-1/2 activation [46]. Therefore, by combining data from our predictive STRING/KEGG and STITCH models (Figure 2 and Figure 4) and our experimental validation experiments (Figure 3), we identified JNK-1/2, p38, JIP-1, and JUN as the main downstream targets responsible for the observed BK channel-mediated inhibition of TNF-α-induced CCL-2 secretion. Since the activation of all four targets (JNK-1/2, p38, JIP-1, JUN) is thought to be a Ca^2+^-dependent process [47,48,49], these findings align well with the inhibitory effects of Nifedipine on CCL-2 secretion (Figure 1).

This is the first report proposing endothelial BK channels as potential therapeutic targets to counteract TNF-α-induced inflammation in the lung. In other organ systems, the modulation of these channels has already sparked clinical interest [50]. NS1619 has been reported to induce an anti-inflammatory phenotype and the mitigation of post-ischemic mucosal barrier disruption in the small intestine by a mechanism that involves reactive oxygen species (ROS)-dependent heme oxygenase-1 (HO-1) activity [51]. Furthermore, the preconditioning of mice with NS1619 prevented post-ischemic increases in intestinal TNF-α levels [51], and in vascular smooth muscle cells, BK channel activity promotes vasodilation and prevents arterial hypertension [52].

The limitations of our study include the need for the validation of the reported anti-inflammatory effects in vivo. It is also important to recognize that, in addition to the plasma membrane, BK channels can also be found on nuclear [53] and mitochondrial membranes [54,55]. Therefore, a potential BK-mediated protective effect in the nucleus and other cellular organelles needs to be further investigated. In addition, given the diverse effects of NS1619 in mitochondria of different tissues [56,57,58], the observed effects need to be investigated in mitochondria from pulmonary endothelial and alveolar epithelial cells. Another more general limitation is the lack of a 100% specificity of virtually all known ion channel modulators, and known dosing variations among cell types, species, and culture conditions. NS1619 is no exception to these limitations, but is still the best-known and most-studied BK channel activator in the literature [18,59,60,61,62]. Although BK-deficient mice are difficult to breed given their fertility problems, in future studies, we will attempt to validate our findings with NS1619 in vivo using BK-KO mice [63], where the effects of NS1619 should be absent.

In conclusion, our data show for the first time the importance of Ca_V_ channels and the role of Em hyperpolarization in TNF-α-induced inflammatory mediator secretion from human pulmonary endothelial cells via Ca^2+^-dependent JNK-1/2 and p38 pathways. These findings lay the groundwork for future studies targeting these pathways and signaling molecules as potential new approaches against TNF-α-induced endothelial inflammation.

## 4. Materials and Methods

### 4.1. Cells

Human pulmonary microvascular endothelial cells (HULEC5a; ATCC-CRL-3244) were purchased from ATCC (Manassas, VA, USA) and cultured in an MCDB131 medium (Calsson Labs, Smithfield, UT, USA) supplemented with 10% FBS (Gibco, Waltham, MA, USA), 1% Penicillin/Streptomycin (Gibco, Waltham, MA, USA), 10 mM L-Glutamine (Gibco, Waltham, MA, USA), 10 ng/mL Epidermal Growth Factor (GenScript, Piscataway, NJ, USA), and 1 µg/mL hydrocortisone (Alfa Aesar, Waltham, MA, USA). Primary human pulmonary artery endothelial cells (HPAEC) were purchased from Lifeline Cell Technology (Carlsbad, CA, USA) and cultured in an endothelial cell medium (ScienCell, Carlsbad, CA, USA), supplemented with 5% fetal bovine serum (FBS), 1% endothelial cell growth supplement, and 1% antibiotic solution (Penicillin/Streptomycin) (ScienCell, Carlsbad, CA, USA). Primary human alveolar epithelial cells (HPAEpiC) were purchased from ScienCell (Carlsbad, CA, USA) and cultured in a human epithelial cell medium (Cell Biologics, Chicago, IL, USA), supplemented with the epithelial cell medium supplement kit (Cell Biologics, Chicago, IL, USA). HULEC5a cells were used at a passage number <20, and HPAEC or HPAEpiC at a passage number <15. Separate passages from a given lot were considered biological replicates.

#### 4.1.1. Cell Culture Models and Cytokine Measurements

CCL-2 and IL-6 concentrations in cell culture supernatants were quantified by ELISA (BD Biosciences, San Jose, CA, USA) after the treatment of cells with 5 ng/mL or 25 ng/mL TNF-α (R&D Systems, Boston, MA, USA; *E. coli*-derived), NS1619 (30 μM; Millipore Sigma, Carlsbad, CA, USA), Nifedipine (10 μM; Alomone Labs, Jerusalem, Israel), JNK inhibitor VIII (20 μM; Millipore Sigma, Carlsbad, CA, USA), SB203580 (20 μM; Millipore Sigma, Carlsbad, CA, USA) or equimolar vehicle controls for 24 hrs. To avoid any potential cell toxicity related to TNF-α, in a pilot dose–response experiments, we treated HULEC5a cells with 5 and 50 ng/mL TNF-α for 24 hrs, measured CCL-2 and IL-6 concentrations in cell supernatants, and assessed cell viability (XTT Cell Viability Assay Kit, Biotium, Freemont, CA, USA).

#### 4.1.2. Em Measurements

Em changes were determined using fluorescence imaging plate reader (FLIPR) Membrane Potential Assay kits (Molecular Devices, San Jose, CA, USA) [1,2,3]. The relative fluorescence of dye-loaded cells was recorded every 7 s using a Synergy-2 Multi-Mode Microplate Reader (Biotek, Winooski, VT; excitation 525 nm, emission 590 nm, at 37 °C) after adding TNF-α (5 ng/mL), NS1619 (30 µM), TNF-α (5 ng/mL) + NS1619 (30 µM), K2SO4 (45 mM), or equimolar vehicle controls for 5 min at at 37 °C after a 1 min recording of the stable background fluorescence. In this assay, a decrease in relative fluorescence represents hyperpolarization, whereas an increase represents depolarization.

To determine the resting plasma membrane potential (Em), we measured baseline Em values using the whole-cell patch clamp technique [64,65].

#### 4.1.3. Gene Expression

The total RNA was isolated from the control and TNF-α-treated (5 ng/mL for 24 h) HULEC5a and HPAEpiC cells, using a Qiagen RNeasy Mini Kit (Hilden, Germany), 1–8 μg RNA were reverse transcribed with a High Capacity cDNA Reverse Transcription kit (Applied Biosystems, Foster City, CA, USA), and 2 μL cDNA were amplified by semi-quantitative Real-Time PCR (TaqMan) with primers for specific *KCNMA1* (BK-α1), or *GAPDH* (Applied Biosystems).

### 4.2. Prediction of Protein–Protein Interaction (PPI) Networks Using Functional Enrichment Analysis (STRING)

To predict the PPI networks underlying the interactions between BK channels, L-type Ca_V_ channels, TNF-α, and CCL-2, we employed a search tool for the retrieval of interacting genes/proteins (STRING; http://string-db.org, accessed on 15 April 2022, version 11.0). STRING integrates all known and predicted associations between proteins, including both physical interactions and functional associations. Furthermore, STRING collects and scores evidence from a number of sources such as (i) automated text mining from the scientific literature, (ii) databases of interaction experiments and annotated complexes/pathways, (iii) computational interaction predictions from co-expression and from conserved genomic context, and (iv) the systematic transfers of interaction evidence from one organism to another. We developed a predicting strategy that included 3 main steps: first, we defined which downstream networks were regulated by BK and Ca_V_ channels. For this purpose, our experimentally validated source proteins (BK channels, Ca_V_ channels, TNF-α, CCL-2) were entered into STRING as source proteins. By setting the confidence of the interaction cutoff level to “high” (0.8, range 0–1), STRING analysis revealed an additional top 9 downstream proteins connected to each other and/or to the experimentally validated initial cluster consisting of BK channels, Ca_V_ channels, TNF-α, CCL-2. Second, we performed “Cluster analysis” using the KMEANS clustering algorithm. This step defines the number of sub-clusters (i.e., subdivisions of the initial network/cluster) within the network predicted in step 1. Understanding the way these sub-clusters are connected forms the basis for step 3, where we used the Kyoto encyclopedia of genes and genomes (KEGG) pathway enrichment analysis in STRING to narrow down the candidate pathways within the network to the top 3. This is the most critical step in defining the most likely downstream signaling pathways employed by BK and Ca_V_ channels during TNF-α-induced inflammation. Those proteins that are common to all 3 different pathways are most likely responsible for the observed BK and Ca_V_ channel-mediated inhibition in CCL-2 secretion [18,37].

### 4.3. Experimental Validation of Predicted PPI Networks

To validate the activation of our top 3 predicted signaling pathways (RIG-1, NF-kB, MAPK), we used an ELISA-based RIG-1-like receptor (RLR) kit (MyBioSource, San Diego, CA; #MBS262313), an NF-kB/p65 (pS536 + Total) detection kit (Abcam; #ab176663), a multiplex immunoassay MAP kinase phosphoprotein (phospho-p38, phospho-ERK-1/2, phospho-JNK-1/2) kit, and an electrochemiluminescent platform (Meso Scale Discovery, MSD, Rockville, Maryland #K15101D-1), respectively [11,66].

### 4.4. Chemical/Drug-Protein Interaction Analysis (STITCH)

To explore the interactions between the chemical NS1619 and target proteins, we input our experimentally defined protein queries (BK channels, Ca_V_ channels, TNF-α, CCL-2, JNK-1/2, and p38) and the chemical NS1619 as entry points into STITCH (www.stitch.embl.de; accessed on 15 April 2022, version 5.0). This database integrates information about interactions from metabolic pathways, crystal structures, binding experiments, and drug–target relationships. Inferred information from phenotypic effects, text mining, and chemical structure similarity is used to predict relations between chemicals and proteins [44]. We defined interactions between the initial cluster and 5 downstream partners, and a “high” interaction score of 0.7 (range 0–1) was considered significant. KEGG pathway enrichment analysis in STITCH was used to define the top candidate signaling pathways.

## 5. Significance Statement

The inflammatory mechanisms that cause damage to the alveolar–capillary interface during acute lung injury (ALI) are poorly understood, and no targeted approaches exist that improve patient outcomes. TNF-α levels are consistently elevated in ALI patients, but the role of Ca^2+^ influx in TNF-α-mediated cytokine secretion remains a matter of intense discussion. Using a TNF-α-induced inflammatory model, we discovered that inflammatory CCL-2 and IL-6 secretion form of human pulmonary endothelial cells relies on voltage-gated Ca^2+^ (Ca_V_) channel activity. We identified cell membrane potential hyperpolarization via the activation of large conductance K^+^ (BK) channels as a new alternative approach to inhibit CCL-2 secretion, which occurs via the BK-mediated downstream inhibition of Ca^2+^-dependent JNK-1/2 and p38 signaling. These findings represent a potentially new targeted approach against TNF-α-induced inflammation in the lung.

## Figures and Tables

**Figure 1 ijms-24-04087-f001:**
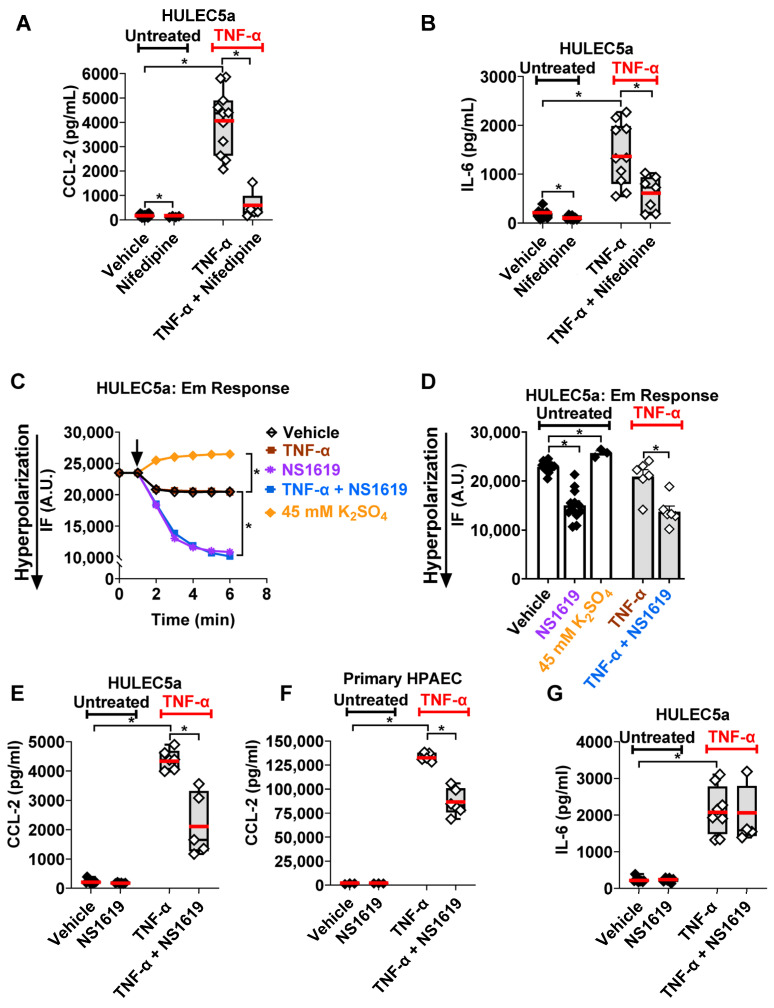
Effects of Ca_V_ channel inhibition with Nifedipine and the selective BK channel opener NS1619 on inflammatory cytokine secretion: (**A**,**B**). The inhibition of L-type Ca_V_ channels with Nifedipine (10 µM; 24 h) decreases the baseline and TNF-α-induced (5 ng/mL; 24 h) CCL-2 and IL-6 secretion from HULEC5a cells (*n* = 5–20, * *p* ≤ 0.05), shown by ELISA. (**C**,**D**) Fluorescence intensity (IF; A.U. = arbitrary units) measured by fluorometric FLIPR assays depicts Em changes in HULEC5a cells: (**C**) A representative recording, showing NS1619-induced (30 µM; 5 min) Em hyperpolarization (decrease in IF), which persists even after TNF-α treatment (5 ng/mL; 5 min). TNF-α itself did not alter the Em. As an internal control, we induced Em depolarization with 45 mM K_2_SO_4_. (**D**) Summary of *n* = 6 individual FLIPR experiments; * *p* ≤ 0.05. (**E**,**F**) Em hyperpolarization with NS1619 (30 µM; 24 h) decreases CCL-2 secretion from human pulmonary microvascular endothelial (HULEC5a) and primary human pulmonary artery endothelial (HPAEC) cells when stimulated with 5 ng/mL or 25 ng/mL of TNF-α, respectively, shown by ELISA. (**G**) Em hyperpolarization with NS1619 (30 µM; 24 h) does not change TNF-α-induced (5 ng/mL; 24 h) IL-6 secretion from human pulmonary endothelial cells (HULEC5a), shown by ELISA. Baseline CCL-2 and IL-6 secretion are not affected by NS1619 treatment (*n* = 4–11, * *p* ≤ 0.05). *n* = number of separate cell passages, which are considered biological replicates.

**Figure 2 ijms-24-04087-f002:**
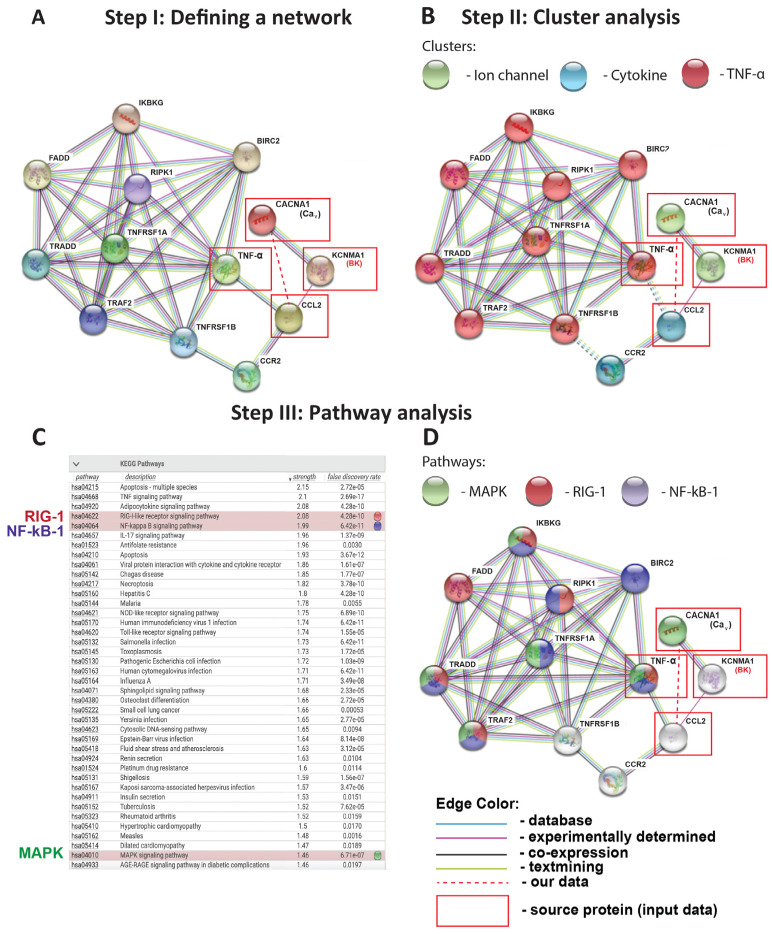
Protein-protein interaction (PPI) prediction model (STRING): (**A**) By entering KCNMA1 (BK channel), CACNA1 (Ca_V_ channel), TNF-α, and CCL2 as query proteins (in red boxes), the STRING PPI network model reveals the nine most likely downstream interaction partners with the confidence level set to “high” (0.8 on a 0–1 scale). (**B**) STRING model of cluster analysis using K-MEANS algorithm to categorize each protein molecule into a determined sub-cluster: Ion channel, Cytokine, TNF-α pathway. (**C**,**D**) KEGG pathway analysis in STRING, highlighting relevant pathways in red (RIG-1), blue (NF-kB-1), and green (MAPK). The “strength” reflects the score of integrated, publicly available PPI information: the higher the “strength” number, the more likely this pathway is present in the model. “False discovery rate” (FDR) is based on a statistical approach used in high-throughput experiments to correct for random events that falsely appear significant: the lower the FDR, the more likely a pathway is involved in a given network. All proteins in this figure are labeled with their gene names, as is conventional in this field. The red dashed line depicts the association between Ca_V_ channels and CCL-2 identified in this study.

**Figure 3 ijms-24-04087-f003:**
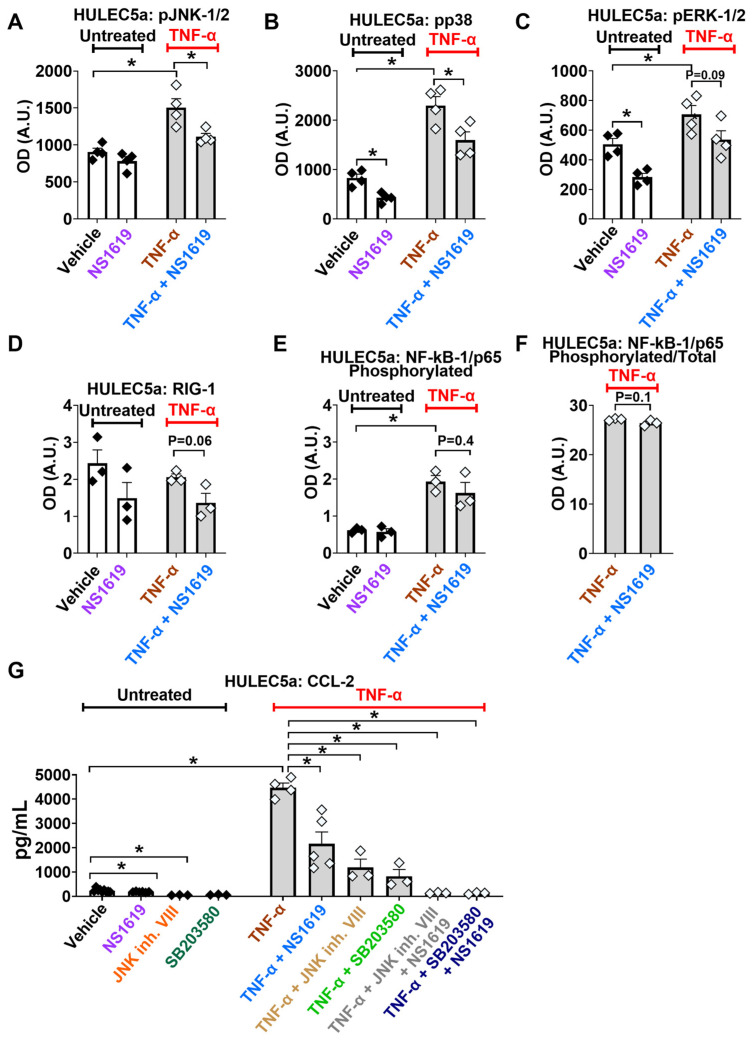
BK channel activation inhibits TNF-α-induced MAPK but not NF-kB-1/p65 or RIG-1 pathways. The treatment of HULEC5a endothelial cells with the BK channel opener NS1619 (30 µM, for 24 h) inhibits the TNF-α-induced (5 ng/mL; 24 h) phosphorylation of the Ca^2+^-dependent JNK-1/2 (**A**) and p38 pathways (**B**), but has no effect on RIG-1 expression or the phosphorylation of ERK-1/2 and NF-kB/p65 (**C**–**F**) (*n* = 3–4, * *p* ≤ 0.05) as shown by ELISA-based multiplex assays. (**G**) The inhibition of JNK-1/2 (JNK inhibitor VIII, 20 µM, 24 h) and p38 (SB203580, 20 µM, 24 h) pathways, with or without NS1619, decreases TNF-α-induced (5 ng/mL; 24 h) CCL-2 secretion from HULEC5a cells (*n* = 3–4, * *p* ≤ 0.0001), shown by ELISA. *n* = number of separate cell passages, which are considered biological replicates.

**Figure 4 ijms-24-04087-f004:**
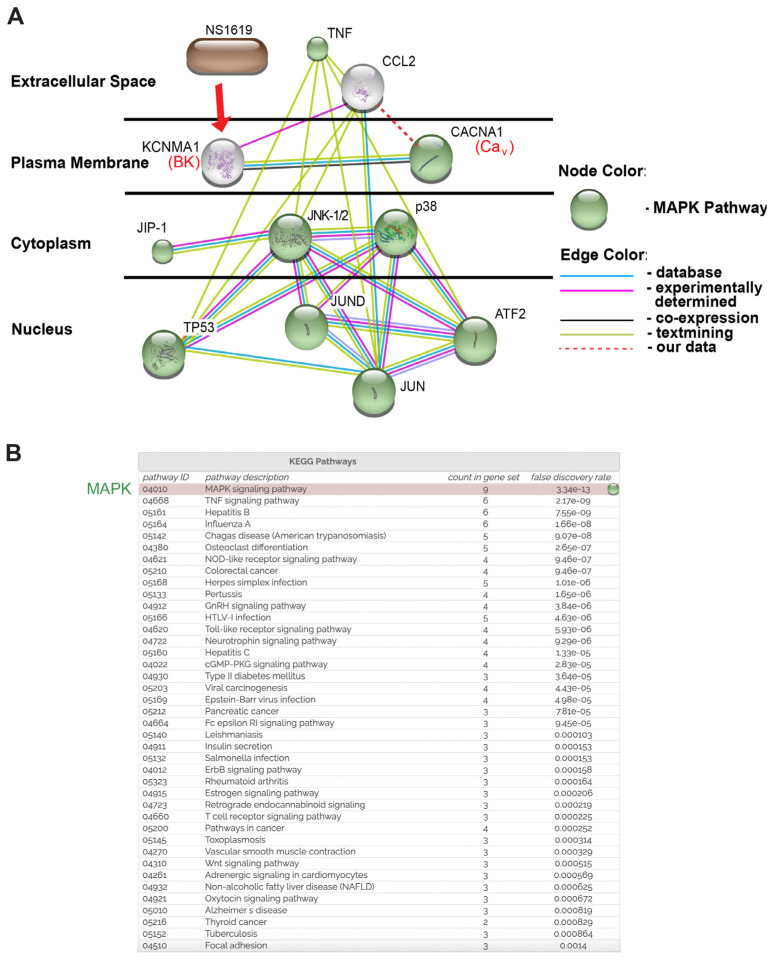
Chemical/drug-protein interaction analysis (STITCH): This putative model predicts that NS1619 reduces TNF-α-induced CCL-2 secretion through the inhibition of JNK-1/2, p38, and JUN activation. (**A**) CACNA1 (Ca_V_ channels), KCNMA1 (BK channels), CCL-2, TNF-α, JNK-1/2, p38, and NS1619 were entered as query parameters along with the predicted five most likely interaction partners: JIP-1, TP53, JUN, JUND, AFT2. The confidence level was set as “high” (0.8 on a 0–1 scale). Proteins are allocated according to their cellular location. Proteins involved in the Ca^2+^-dependent MAPK pathway are indicated in green. All proteins in this figure are labeled with their gene names, as is conventional in the literature. The red dashed line depicts the association between Ca_V_ channels and CCL-2 identified in this study. (**B**) KEGG pathway analysis of the STITCH model highlights the MAPK pathway as the top candidate pathway. “Count in gene set” is a parameter in gene enrichment analysis: a powerful analytical method for interpreting gene expression data. It ranks all genes in a data set, then calculates an enrichment score for each gene set (the higher the “count”, the more often members of that gene set occur at the top of the ranked data set).

**Figure 5 ijms-24-04087-f005:**
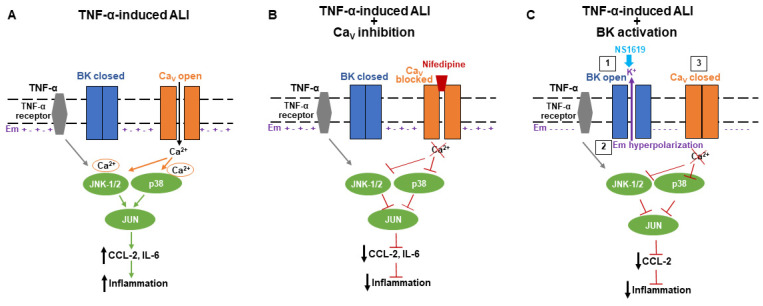
Proposed schematic diagram of Ca_V_ and BK channel-mediated regulation of CCL-2 secretion during TNF-α-induced inflammation: (**A**) Given the relatively depolarized Em of endothelial cells (indicated as + - + -), the open probability for Ca_V_ channels is high, which promotes Ca^2+^ influx, the activation of Ca^2+^-dependent MAPK (JNK-1/2, p38, JUN) signaling, and downstream CCL-2 and IL-6 secretion and inflammation. (**B**) Inhibition of L-type Ca_V_ channels with Nifedipine prevents Ca^2+^ influx-mediated MAPK activation and decreases downstream CCL-2 and IL-6 secretion. (**C**) BK channel opening with NS1619 results in Em hyperpolarization (indicated as - - - -) and the closing of Ca_V_ channels, thus preventing Ca^2+^-dependent MAPK activation and downstream CCL-2 secretion.

## Data Availability

Data are expressed as Box-Whisker blots with mean, max, and min values, or bar diagrams with mean ± SEM. Comparisons between groups were performed using Mann–Whitney U or Student *t*-tests and confirmed by two-way ANOVA analysis (GraphPad 8.3, San Diego, CA, USA), with *p* ≤ 0.05 considered as significant.

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
