# Peer review of "Activation of Endothelial Large Conductance Potassium Channels Protects against TNF-α-Induced Inflammation"

_ijms, 2023, doi:10.3390/ijms24044087_

Round 1

Reviewer 1 Report (New Reviewer)

The manuscript „Activation of Endothelial Large Conductance Potassium Channels Protects against TNF-α-induced Inflammation“ by Tatiana Zyrianova, Kathlyn Zou, Benjamin Lopez, Andy Liao, Charles Gu, Riccardo Olcese, and Andreas Schwingshackl presents new data suggesting that calcium influx mediated by voltage-gated calcium channels is responsible for TNF-α-induced endothelial secretion of CCL-2 and IL-6.

This referee has the following comments:

  1. General comment: In this study a cultured human pulmonary microvascular endothelial cell line and cultured primary human pulmonary artery endothelial cells have been used. Of note, cell culture is known to affect gene expression considerably. Consequently, findings obtained from cultured cells cannot be extrapolated to these cells in vivo without evidence that the observed findings can be obtained in non-cultured cells, intact organs or in vivo. Thus, in all sections of the manuscript it should be indicated whether reported findings have been obtained from cultured or non-cultured cells and extrapolation of findings obtained from cultured cells to in vivo function should be avoided.  
  2. Introduction: Knowledge gaps are described, but the hypotheses to be tested are missing. Please add the hypotheses that you have been tested to the introduction section.
  3. Results, p.3, para2: It is not correct to state that whole cell measurements of the membrane potential provide information about the open state probability of Cav channels. The open state probability can only be obtained using single channel patch-clamp recordings. What can be estimated is the role of Cav channels for setting the membrane potential. However, this was not achieved as the measured depolarized membrane potential may be caused by a high permeability for Cav cannels but may also be caused by a high permeability for any other depolarizing current, especially TRP channels shown to be expressed in endothelial cells. Thus, in order to be able to assess the contribution of Cav channels to the membrane potential, a considerable effect of nifedipine on the membrane potential should be shown or Cav currents be measured in the whole-cell mode of the patch-clamp technique. Please provide such data or adjust the interpretation of the data. Accordingly, the discussion p.6, last para should be revised.
  4. Results, p.3, para3: The statement “Based on our findings with Nifedipine showing that inflammatory CCL-2 and IL-6 secretion can be inhibited by blocking Ca2+ influx via endothelial L-type CaV channels,…” is not supported by the data. The data in Fig. 1A,B just “demonstrate a role for L-type CaV channels in these TNF-α-induced secretory processes” (as you stated in para1). The conclusion that this role of Cav channels is related to Ca2+ influx via endothelial L-type CaV channels has not been shown as Ca influx was not measured. Alternatively, nifedipine may simply lower basal calcium required as a sort of threshold for the TNF-induced signaling or lower the amount of calcium in calcium stores that may be involved in TNF signaling.    
  5. Results, p.3, para 3: Unfortunately, NS1619, especially at the high concentration used, is not a selective BK channel opener. It acts on calcium and potassium channels (e.g. Holland, Langton, Standen & Boyle, 1996). Thus, either more selective BK channel opener should be used (e.g. NS19504) and their effect should be sensitive to the specific BK channel inhibitor iberiotoxin, or the interpretation of the data should be modified.
  6. Results, p.3, para4: The second reference to Fig. 1E probably means Fig. 1F.    
  7. Results, p.5, para1: The statement “Although NS1619 decreased RIG-1 expression in unstimulated and TNF-α-stimulated endothelial cells, these effects fell short of meeting statistical significance.” is statistically incorrect. If p>0.05, the statistical threshold you selected, the data presented show that you have not been able to detect an effect. An effect that does not meet significance does not exist. Moreover, since with n=3 statistical analysis is problematic, these data should be considered preliminary.
  8. Results, p.5, para2: The statement “These findings imply a role for JNK1/2 and p38 signaling in the NS1619-mediated effects.” is unclear. First, the inhibitors reduced basal secretion. May be this alone is sufficient to reduce the effect of TNF on secretion and the inhibited pathways have nothing to do with the effect of TNF itself? Second, if the inhibited pathways mediate the effect of NS1619, as suggested, it is expected that the effect of NS1619 is abolished in the presence of the inhibitors. This seems not to be the case, NS1619 seem to have an additional effect. Please clarify.
  9. Results, p.5, para 3: Here the confidence level of interaction is given with 0.7, in the legend to figure 4 it is 0.8. Please revise.
  10. Discussion, p.6, para 2: The statement “Therefore, our Nifedipine studies show for the first time the functional expression of L-type CaV channels in human pulmonary microvascular endothelial cells.” is not correct. First, nifedipine at the concentration used has well known effects on other signaling molecules. Second, the patch-clamp experiments provided do not support the existence of Cav channels, see comment above. Finally, since your data have been obtained on cultured cells and because cell culture is known to affect gene expression considerably, there is no evidence, that human pulmonary microvascular endothelial cells express these channels.
  11. Methods: The link to Stitch does not work.
  12. Methods, statistics: Please state what n (number of observations) means in your study, biological or technical replicates. This is important because the use of multiple cells from one cell line can be considered only as n=1, if n is biological replicates. In addition, n=3 is too small to calculate firm statistics. Such data should be considered and marked as preliminary.

Author Response

Reviewer 2 Report (New Reviewer)

The present work shows the role of Ca2+ influx in TNF-α-mediated inflammation in human pulmonary microvascular endothelial cells. The authors evaluated the participation of L-type voltage-gated Ca2+ (CaV) channels and large conductance K+ (BK) channels in this process. I have a few remarks and comments on the work.

General comment:

Firstly, the article is not designed according to the IJMS template, the authors need to show respect for the editors and reviewers and follow the recommendations of the journal.

Comments on results and discussion.

The main problem of this manuscript is an analysis of multiple experimental groups by Mann-Whitney U or Student t tests instead of ANOVA, because some of the indicated significances may not hold up to testing among all experimental groups.

In the third paragraph of the results, the authors write that they used the selective BK channel opener, NS1619 (30 µM). Unfortunately, the NS1619 isn't that selective, and that's something to keep in mind. In particular, this agent is able to influence mitochondrial respiration, etc., which has been shown in in vitro and in vivo studies (see PMID: 12781334, PMID: 20021124, PMID: 36365155). The authors need to more carefully interpret the results according to this remark and rewrite the corresponding sentences.

In addition, it is important to understand that BKCa channels are also found in nuclear and mitochondrial membranes. It is necessary to think about the contribution of channel activation in these organelles to the results of the work, or to state the corresponding limitation of the study.

Author Response

Reviewer 3 Report (New Reviewer)

The manuscript by Zyrianova et al sought to investigate the role of calcium influx in TNF-a-induced chemokine/cytokine release by lung endothelial cells and the mechanism of inflammatory cytokine secretion via activation of the BK channel. This is an interesting and well-performed study. The reviewer has the following comments.

  1. What might cause the drop in Em or towards Em hyperpolarization in Figure 1 C and D. Although the degree of change may not be statistically significant as stated by the author, it is a quite big decrease in Em. Should TNF-a cause Em depolarization because TNF-a can activate the L-type Cav channel and cause Ca2+ influx?
  2. The study used HULEC or HPAEC to study the mechanism of TNF-a-induced inflammatory cytokine secretion. Will these drugs also produce similar results in vivo or related animal models, such as the acute lung injury model? Will these treatments reduce cytokine release in the animal or co-culture and reduce leukocyte recruitment, adhesion, and infiltration? Or at least, in patients taking Nifedipine, is there any indication of reduced pulmonary inflammation?
  3. Will gene knockdown of the L-type Cav channel or overexpression of the BK channel give similar results?
  4. In the figure legend Figure 1, there are two (G).

Round 2

Reviewer 1 Report (New Reviewer)

The manuscript „Activation of Endothelial Large Conductance Potassium Channels Protects against TNF-α-induced Inflammation“ by Tatiana Zyrianova, Kathlyn Zou, Benjamin Lopez, Andy Liao, Charles Gu, Riccardo Olcese, and Andreas Schwingshackl has been revised.

This referee has the following comments:

  1. Abstract: The addition of the hypothesis to the introduction section is appreciated. However, the hypothesis should also be presented in the abstract.
  2. Results, whole cell measurements: The data announced in your response seem not be present in the manuscript, please check.
  3. Results, NS1619: The reference to your using the more specific BK channel opener NS19504 is appreciated. Unfortunately, even NS19504 is not specific, at least at the high concentrations used in your study (see Ma D et al 2020, Supplement). Thus, you should show that your effects of NS1619 or of NS19504 can be reproduced at concentrations reported in the literature to be inhibited completely by specific BK channel inhibitors or, even better, you should demonstrate that the effect induced by the BK channel opener in your study is blocked by a specific BK channel inhibitor (IBTX, paxillin).
  4. Methods, statistics: Your clarification regarding what n means is appreciated. However, it should be clarified in more detail why you consider different cell passages as biological replicates. When cells in different passages are derived from the same predecessor, passages are often considered just as technical replicates.

Author Response

Reviewer 2 Report (New Reviewer)

The authors improved the presentation of the manuscript and provided adequate answers to questions. A small remaining note: I still recommend adding information about the pleiotropy of NS1619 (my third comment). This is important for people who are getting acquainted with the topic and action of this agent for the first time.

Round 3

Reviewer 1 Report (New Reviewer)

no comment

This manuscript is a resubmission of an earlier submission. The following is a list of the peer review reports and author responses from that submission.

Round 1

Reviewer 1 Report

The work entitled “Activation of endothelial Large Conductance Potassium Channels Protects against TNF-alpha-induced Inflammation” by Zyrianova, et al. is an interesting investigation of the impact of CaV and BK channels on human pulmonary microvascular endothelial cells on TNF-alpha induced chemokine and pro-inflammatory cytokine production. The authors combine multiple techniques to develop a very nice model to explain their findings. However, there are concerns with the document in its current form. Please see below for specific comments.

  1. The authors use only one human pulmonary microvascular endothelial cell line to make their conclusions. There is a risk that this finding is specific to the particular cell line that they are using. Other human pulmonary microvascular endothelial cell lines are commercially available. A more decisive conclusion requires experiments that demonstrate the two drugs they used on the HULEC5a cell line induced a similar finding (as shown in figure 1) in at least one other human pulmonary microvascular endothelial cell line.
  2. The authors conclude “the BK channel activation-mediated decrease in TNA-alpha-induced CCL-2 secretion is caused by inhibition of Ca2+ influx-dependent JNK-1/2 and p38 signaling pathways.” The authors do not definitively show that the decrease in CCL-2 is due to activation of these pathways. It is just a correlation. If this is true, then inhibitors of these pathways would block the decrease in CCL-2 in the presence of TNF-alpha and the BK channel opener, NS1619. To make this definitive statement, the experiment that supports it needs to be performed. Otherwise, ensure the statements say that the findings correlate but do not cause.
  3. This work is almost identical to a recent article from this group that analyzes the impact of LPS on the same cell lines (PMID: 33217242) that also analyzes CCL2 and IL-6 production by the HULEC5a cell line. It is somewhat surprising that the authors barely comment on this work, since LPS is a traditional inducer of Acute Lung Injury (ALI) in vivo. The authors describe their in vitro model as an “ALI model” but this in vitro model is physiologically very different from a true in vivo ALI model. The overuse of the term “ALI model” when the study does not address actual lung injury throughout the literature confuses readers. Please refrain from using this term throughout the document. It is fine to project how the current findings may impact an in vivo model, but without doing an in vivo experiment with lung injury, it is incorrect to publish it as an “ALI model”. Finally, please add a discussion of this previous work into the discussion of the current manuscript to address the similarities found between previous and current works.

Reviewer 2 Report

Dear Authors Unfortunately I am rejecting this article since there are serious problems regarding the experimental design, the conclusions and the presention of results and method. Specifically:

In Figure 1 but mostly everywhere they do not specify the type of Experiment (Elisa) they performed.

It is not clear how they choose the target protein to be anlyzed by STRING program.

In figure 3 how they validate the phosphorylation..

Overall the paper has to be re-written. In the present form is impossible to follow and the research design is not appropriate. The conclusion are not based on solid Results.